# Effects of a Fat-Rich Diet on the Spontaneous Release of Acetylcholine in the Neuromuscular Junction of Mice

**DOI:** 10.3390/nu12103216

**Published:** 2020-10-21

**Authors:** Carlos Gimenez-Donoso, Marc Bosque, Anna Vila, Gemma Vilalta, Manel M Santafe

**Affiliations:** 1Centre de Fisioteràpia Inspira, Carrer Muntaner num 200, 2º, 2ª, 08036 Barcelona, Spain; carlos.gimenez@centroinspira.es; 2Unit of Histology and Neurobiology, Department of Basic Medical Sciences, Faculty of Medicine and Health Sciences, Rovira i Virgili University, Carrer St Llorenç num 21, 43201 Reus, Spain; fisioterapia.marc@gmail.com (M.B.); annaviimo@gmail.com (A.V.); gemmaspirit1@gmail.com (G.V.)

**Keywords:** electromyography, high-fat diet, myofascial pain syndrome, obesity, spontaneous neurotransmission

## Abstract

Western societies are facing a clear increase in the rate of obesity and overweight which are responsible for musculoskeletal pain. Some of the substances described in the environment of myofascial trigger points (MTrPs) are the same as those found in the skeletal muscle of obese people, such as cytokines. Furthermore, elevated neuromuscular neurotransmission has been associated with MTrPs. The main objective of this study is to assess whether obesity or overweight may be a facilitator of myofascial pain. The experiments were performed on male Swiss mice. One experimental group was given a typical “cafeteria” diet and another group a commercial high-fat diet for six weeks. Intramuscular adipocytes were assessed with Sudan III. The functional study was performed with electromyographic recording to determine the plaque noise and intracellular recording of miniature endplate potentials (MEPPs). The intake of a cafeteria diet showed the presence of more adipocytes in muscle tissue, but not with the fat-supplemented diet. Both experimental groups showed an increase in the plaque noise and an increase in the frequency of MEPPs that lasted several weeks after interrupting diets. In summary, the supply of a hypercaloric diet for six weeks in mice increases spontaneous neurotransmission, thus facilitating the development of MTrPs.

## 1. Introduction

At present, Western societies are having a clear increase in the rate of obesity and overweight [1]. From 1975 to 2016, the global obesity rate has tripled [2]. By 2030, over 38% of the world’s adult population will be overweight and 20% will be obese [3]. Obesity and overweight are considered an epidemic related to the development of various pathologies such as diabetes, cardiovascular disease, metabolic syndrome and musculoskeletal pain [3]. 

There seems to be a strong relationship between obesity and pain [4]. Usually, the association between overweight and musculoskeletal pain has been attributed to an increase in the mechanical stress caused by overweight in load bearing joints. However, the literature also shows an association between pain and overweight in joints that do not support load [3,5,6,7,8,9]. For example, associations between overweight and symptomatic osteoarthritis of the hand [5], shoulder and neck pain [6], the number of episodes and intensity of migraine attacks [7,8], even with neuropathic pain [9] have also been described. Thus, the relationship between overweight and musculoskeletal pain appears to be at least, in part, independent of the mechanical overload and probably involves systemic phenomena.

Obesity is accompanied by a chronic inflammatory response with high production of adipokines (IL-6, TNF-α, adiponectin, leptin, and resistine) and macrophage infiltration into the adipose tissue [10]. This chronic inflammatory response has been linked to metabolic syndrome and musculoskeletal pain [11]. Moreover, musculoskeletal studies demonstrate the impact of these cytokines on muscle metabolism [12]. However, there are no studies that analyze the impact that obesity may have on other muscle functions or on muscle pain.

Myofascial pain syndrome (MPS) is the set of sensory (local and referred pain), motor (limited range of motion, weakness) and autonomic signs and symptoms caused by myofascial trigger points (MTrPs) [13]. The prevalence in the general population ranges from 20% to 90% and it is suggested that practically, all adults will suffer at least one episode of myofascial pain in their lifetime [14].

MTrPs have been proposed to be a neuromuscular dysfunction in which abnormal motor end plate function results in an excessive concentration of acetylcholine (ACh) in the synaptic cleft [15]. This excess ACh causes a localized contraction of the sarcomeres below the neuromuscular junction. Thereupon, a cascade of events that cause cellular suffering, local acidic pH, and the release of sensitizing substances from the nociceptive nerve terminals occur [16]. Altogether these changes are responsible for local pain. In 2005, Shah identified a high presence of several of these substances (pH, SP, IL-6, TNF-α, CGRP…) in the environment of MTrPs [16]. On the other hand, in our laboratory, we experimentally cause the appearance of contracted subsynaptic sarcomere by increasing local ACh with anticholinesterase drugs [17]. This increase in ACh release can be recorded by needle electromyography as spontaneous low-voltage electrical activity (30–60 mV) like endplate noise [14,17,18].

Some of the substances that Shah [16] found in the environment of active MTrPs are the same as those found in the skeletal muscle of obese people: IL-6, IL-1B, TNF-α [12]. No one has ever established a relationship between obesity and MPS before. However, given that in both clinical situations, a similar inflammatory profile surrounding muscle tissue is found, we hypothesized that muscle neurotransmission in obese or overweight individuals is increased, thus constituting a predisposing factor for the development of MTrPs. 

In the present study, intracellular recordings and electromyography were performed at the end of the diet exposure period and several weeks after. An increase in the body weight of the mice was paralleled by a significant increase in the spontaneous release of acetylcholine that lasts for several weeks after diet manipulations. All the results obtained suggest that obesity and overweight can cause myofascial muscle pain.

## 2. Materials and Methods

The mice were cared for in accordance with the guidelines of the European Community’s Council Directive (2010/63/EU) and the Spanish Royal Decree 53/2013 for the humane treatment of laboratory animals. The Animal Research Committee of the Universitat Rovirai Virgili (Reference number: 0233) reviewed and approved all experiments on animals. The experiments were performed on young (45–50 days) adult Swiss male mice (Charles River, L’Arbresle, France). Mice were habituated to the facility for at least 1 week prior to studies and were housed in groups of four, with sawdust bedding and ad libitum access to water and food throughout the study. The animals’ rooms were maintained at a temperature of 22 ± 2 °C, a relative humidity of 50 ± 10%, and a 12 h light/dark automatic light cycle.

### 2.1. Animals and Dietary Protocol

Mice were randomly divided into cages of 2 animals. The cages were randomly grouped into three groups (see Figure 1): a control group (CTR; N = 10 cages, 20 animals), a “cafeteria diet” group (CAD; N = 10 cages, 20 animals) and a high-fat diet group (HFD; N = 10 cages, 20 animals). During the experiment, the CTR group received a normal laboratory diet (SAFE Diets: 230 HF Rat & Mouse Diet, Augy, France). A “cafeteria diet” model (detailed below) was added to the CAD group and a fat-supplemented diet (230 HF Rat & Mouse Diet, SAFE, Augy, France) was added to the HFD group [19]. Exposure to this type of diet was carried out during a period of 6 weeks. After 6 weeks, both the cafeteria diet and the high-fat diet were withdrawn and all mice were fed exclusively with the regular rodent chow for 3 extra weeks. At the end of the diet exposure (6 weeks), 4 animals were sacrificed for histological studies and 4 animals for electrophysiological and electromyographical recordings. Then, during the next three weeks after diet exposure, 4 animals were sacrificed each week for electrophysiological and electromyographical recordings (Figure 1).

The diets supplied were:

All animals were provided with regular rodent chow (SAFE A04 diet, Panlab, Barcelona, Spain) ad libitum. The composition of this diet was shown in Table 1.

The cafeteria diet (CAD) used in this study consisted of industrial pastries rich in saturated fat (cakes and cookies filled with chocolate) and fried peanuts [19,20,21,22,23]. The average composition is shown in Table 1. Every 2 days, the chopped pastries and peanuts were introduced together in the cage for the animals to eat *ad libitum*. Each time, the old cafeteria diet leftovers were removed and a new ration supplied. At all times, the animals had free access to their usual feed, so the animals continued to ingest the necessary nutrients so as not to suffer any nutritional deficiency. To ensure the amount of cafeteria diet eaten by the mice, the food introduced into the cage was weighed and the food debris was reweighed when removed. Each cage of 2 animals consumed 56 g of the CAD diet per week.

The high-fat diet (HFD), the other type of diet used in this study to induce overweight in animals was a diet enriched in fat (SAFE Diets: 230 HF Rat & Mouse Diet, Augy, France). Unlike the cafeteria diet, this diet is free of additives, colorants, stabilizers and flavorings that could interfere with the results [24,25,26]. The composition of this diet is shown in Table 1. This type of diet was placed inside the cage to facilitate its availability. As with the cafeteria diet, the food was weighed before placing it in the cage and when removing the remains in the next supply. At all times, the animals also had free access to their usual feed to ensure that they continued to ingest the necessary nutrients and not incur any nutritional deficiencies. Each cage of 2 animals consumed 60 g of the HFD diet per week.

### 2.2. Muscles

Animals were deeply anesthetized with isoflurane before being euthanized by exsanguination. The *levator auris longus* (LAL) was excised and dissected on a Sylgard-coated Petri dish containing normal Ringer solution (containing (in mM): 135 NaCl, 5 KCl, 2.5 CaCl_2_, 1 MgSO_4_, 1 NaH_2_PO_4_, 15 NaHCO_3_ and 11 glucose) continuously bubbled with 95% O_2_/5% CO_2_. The LAL muscles were used for methylene blue staining and immunologically labeled. The gastrocnemius muscles were used for electromyographic recordings. LAL muscles were used for Sudan III fat staining. The LAL is a small, flat muscle located immediately under the skin of the murine skull and is extremely useful for intracellular recording techniques (to visualize the muscle fibers and localize the possible synapses requires flat, thin, and transparent muscles). It is also useful for histological techniques since it does not require microtomy, thus minimizing the appearance of the artifacts.

### 2.3. Sudan III

This histological classical staining was performed in the LAL muscle of all the experimental groups and the controls at the end of the period of exposure (6 weeks). Sudan III stains lipids orange-red [27]. The LAL muscles were extracted and fixed in formalin.

The LAL muscle is a flat, thin muscle that does not require microtomy. Whole LAL muscles were immersed in the Sudan III preparation (50 mL of 50% alcohol, 50 mL acetone, 1 g Sudan III—Sigma-Aldrich, Steinheim, Germany) for 5 min. After cleaning the excess dye with 50% alcohol, a methylene blue contrast stain was performed (1 min). After washing off the excess under tap water, it was mounted on glycerin for visualization.

### 2.4. Endplate Noise Recordings

Electromyography (EMG) recordings were obtained from an anesthetized animal at controlled room temperature (22 °C–25.8 °C). The muscle used for this study was the gastrocnemius because of its ease of access and suitability. Recordings were obtained with an electromyography system (MedelecMystro plus, GR20) using a monopolar EMG needle (Natus Manufacturing Limited, London, UK) [17]. The needle was slowly inserted into the muscle and once inside, it was moved in order to enable recording in all directions. The muscle was divided into twelve areas to cover both the entire muscle and avoid recording the same endplate noise twice [17]. The recording needle was introduced into the gastrocnemius until an audible change was heard. The electromyography screen was then checked and if correct (without an alternating current, artifacts, etc.), the endplate noise was recorded. The number of areas with endplate noise (maximum twelve) and the frequency (number of potentials per second that appeared, expressed in Hz) were recorded. 

### 2.5. Electrophysiology: Intracellular Recordings

Spontaneous miniature endplate potentials (MEPPs) were recorded intracellularly with conventional glass microelectrodes filled with 3 M KCl (20–40 MΩ). Records were rejected if the membrane potential was <−50 mV or if it fell by more than 5 mV during the recording period. 

The recording electrodes were connected to an amplifier (Tecktronics, AMS02, Tektronix, Inc., Beaverton, OR, US). A distant Ag–AgCl electrode connected to the bath solution via an Agar bridge (Agar 3.5% in 137 mMNaCl) was used as a reference. The MEPPs were digitized (DIGIDATA 1200 Interface, Axon Instruments Inc, San Jose, CA, USA), stored, and analyzed using a computer. The Axoscope 10.2 was used (Axon Instruments Inc.) for data acquisition and analysis. The MEPP frequency was recorded for 100 s from at least 15 different neuromuscular junctions and the mean values were determined. The mean amplitude (mV) per fiber was calculated and corrected for non-linear summation [28], assuming a membrane potential of –80 mV. 

### 2.6. Statistical Procedure

Values are expressed as the mean ± SEM. In some instances, the values are expressed as “percentage of change”. This is defined as: (experimental value/control value) × 100. We used the two-tailed Welch’s *t*-test for unpaired values because our variances were not equal. This test was chosen as it is more conservative than the ordinary *t*-test. Differences were considered significant at *p* < 0.05.

## 3. Results

### 3.1. Body Weight Evolution

All the mice in each of the three groups increased their body weight during the first 6 weeks. However, the two groups supplemented with the hypercaloric diets (CAD and HFD) increased their weight over the values of the control group (Table 2). At the end of the exposure, at 6 weeks, the group of mice subjected to a cafeteria diet increased their body weight by 52% more than the controls and the group of mice subjected to a high-fat diet increased their weight by 45% over the weight of the controls.

By suppressing the supplementation of the cafeteria and the high-fat diets and maintaining the usual rodent chow, both groups reduced their weight from the first week to match the weight of the controls of the same age (Table 2).

It is confirmed that exposing the mice to 6 weeks of either a cafeteria diet or a high-fat diet causes overweight. By eliminating the hypercaloric diet supplementation, the animals reduced the previously acquired overweight by the first week.

### 3.2. Muscle Fat

In the group subjected to a cafeteria diet, a greater amount of adipocytes appeared between the muscle fibers of the LAL (Figure 2B) than in the controls (Figure 2A) and the HFD group (Figure 2C). This technique was applied to four animals per group (control, CAD and HFD).

### 3.3. Electrophysiology: Intracellular Recording

As shown in Figure 3, at the end of the 6 weeks of exposure to the cafeteria diet, a significant increase in the frequency of MEPPs was observed, which is maintained in the following three weeks after withdrawing from the cafeteria diet. On the other hand, at the end of the 6 weeks of exposure to the high-fat diet, a potent increase in the frequency of MEPPs was also observed, greater than in the CAD group (Figure 3). This increase in spontaneous neurotransmission decreased immediately from the first week after withdrawing from the high-fat diet, but remained high for the next 3 weeks.

The size of the MEPPs did not change at any time in any of the experimental groups (Figure 3B). The data in Figure 3 are included in Appendix A (Appendix A).

### 3.4. Electromyography

Upon the hypercaloric diets, the number of areas with plaque noise increased similarly in both the CAD and HFD groups (Figure 4A). When the diet was withdrawn, the number of areas with plaque noise in the CAD group remain elevated for 2 weeks but in the HFD group, it was elevated for only one week.

Regarding the number of events in each area with plaque noise, a similar increase was obtained in the MEPPs record (Figure 4B): it was more powerful in the HFD group than in the CAD group. When withdrawing the dietary supplementation, as with the MEPPs recording, the HFD group returned to the control values faster than the CAD group. However, at 3 weeks, the two groups achieved control values. The data in Figure 4 are included in Appendix A (Appendix A).

In summary, the increase in the release of ACh caused by the CAD and HFD diets tends to last longer in time once the supplementation is withdrawn.

## 4. Discussion

The main hypothesis of this study is that the accumulation of fat in the skeletal muscle of overweight individuals causes an increase in the spontaneous neurotransmission at the neuromuscular junction.

### 4.1. Overweight

Different types of diets useful in achieving overweight rodents have been described [19]. Initially, in the present study, a hypercaloric CAD was used. This type of diet has been shown to produce overweight mice by increasing the accumulation of fat in the different tissues [20]. However, this type of diet contains several substances such as preservatives, colorings, salt, and processed sugars that could have an effect beyond the induction of overweight. In a bid to isolate the effects of being overweight, a second intervention group was used, which was administered as HFD, which has also been shown to induce overweight in mice [24], but which is free of the remaining substances that could interfere with the results. 

Our results confirm that 6 weeks of exposure to both a CAD as used in our study, as well as a HFD, are sufficient to achieve a significant increase in the weight of the mice (50%) compared to individuals of the same age on a normal diet. Therefore, exposure to these diets is a suitable model for studying overweight.

Similarly, the weight gain achieved with both diets is quickly reversed to a normal diet from the first week. In this sense, other studies such as the one by Reynés et al. [21] have shown that it is possible to reverse the overweight caused by the administration of a cafeteria diet for 1.5 months following a normal diet for 1.5 months in rats. 

Furthermore, the rapid weight reduction obtained in the first week of withdrawal from the cafeteria diet in the present study was consistent with the results obtained by other authors such as Lalanza et al. [22] These authors described that after one week of withdrawal from the cafeteria diet, the overweight caused in rats subjected to 2 months of the cafeteria diet was reversed. This rapid decrease in the weight of the animals was justified by Rogers in 1985 [23] by which the change from a more “tasty” diet to a diet based on feed causes severe hypophagia, especially in the first days after the change.

### 4.2. Muscle Fat

Once the weight gain of the mice in both groups was confirmed, it is interesting to see if this weight gain was accompanied by a greater presence of fat in the muscle tissue. The Sudan III technique allows identifying a greater number of adipocytes between the muscle fibers only in the group subjected to a cafeteria diet. However, in the group subjected to the fat diet, despite suffering the same weight gain, no differences were observed in the accumulation of adipocytes in the muscle tissue compared to the control group. Little variations in the diet composition are an important issue to take into consideration as the Cafeteria diet is higher in carbohydrates and levels of proteins are lower than those in the HFD. 

There are previous studies that compare the cafeteria diet with a diet rich in fat for rats and describe, in addition to an increase in weight, an increase in visceral and subcutaneous fat. These studies showed that the changes caused by the cafeteria diet are greater than those caused by the high-fat diet [19,21]. It has been proposed that the cafeteria diet induces a greater hyperphagia in animals than the high-fat diets that cause a stabilization in the amount of daily intake after the first weeks [21]. In the present study, only muscle fat was studied and no assessment of other fat deposits was made. However, both groups significantly increased their weight, which suggests that HFD also caused accumulations of visceral fat, although, it did not infiltrate the muscle tissue. 

Kahn et al. in 2015 [29] described an increased presence of intramuscular fat in mice which are subjected to a high-fat diet. However, these authors maintained the exposure to a high-fat diet for a much longer time—24 weeks—than in the current study. It is possible that to obtain an increase in muscle fat, the HFD group needs a longer exposure time to the diet. Note that in that study [29] an increase in macrophages in the muscle was demonstrated before the muscle fat was increased. In the present study, the study of the presence of fat 3 weeks after stopping the cafeteria diet was not carried out since weight is normalized.

### 4.3. Spontaneous Neurotransmission

There is an increase in spontaneous neurotransmission in mice subjected to both diets, this increase was more important in the HFD group than in CAD. When the diet was withdrawn, the HFD group maintained this increase for more weeks than the CAD group. The alterations registered intracellularly and those registered with EMG do not behave the same since the intracellular recording is a more sensitive technique than EMG and therefore detects more subtle changes.

In the intracellular registry, the amplitude of the MEPPs was not modified in any experiment. This suggests that the possible accumulation of intramuscular fat does not affect the functional integrity of muscle fiber. In this sense, Garcia et al. [30], exposing muscles to anti-motor neuron IgG, injured the axons, but the muscle remains preserved, in such a way as to obtain a pathological increase in MEPPs without any variation of amplitude. 

The supplementation of both diets causes an increase in spontaneous neurotransmission at the neuromuscular junction that persists for a time longer than the exposure to the diet. This result confirms only part of the hypothesis in the present study, which states that high-calorie diets cause an alteration in spontaneous muscle neurotransmission. However, the results obtained do not corroborate a correlation of this increase in neurotransmission with being overweight and the accumulation of muscle fat, since an increase in neurotransmission was found in the HFD despite no accumulation of muscle fat. These results open new questions regarding the mechanism by which an increase in spontaneous muscle neurotransmission is brought about by this type of diet. However, the fact that coherent changes appear in muscles as disparate functional strengthens the hypothesis that the studied effects of these diets on neurotransmission must be systemic in nature.

A possible factor that may explain these results is the involvement of the sympathetic nervous system. In a 2012 review, Smith and Minson [31] stated that there is sufficient evidence to show that increased fat deposits correlate with increased activity of the sympathetic nervous system in certain tissues such as the kidneys or skeletal muscle. Previously, in 1994, Scherrer [32] demonstrated that individuals with a BMI > 27 have a rate of sympathetic discharge to skeletal muscle that is twice as high as in lean individuals. On the other hand, Chen et al. [33] demonstrated that the activity of the sympathetic nervous system blocked by phentolamine may decrease the spontaneous neurotransmission recorded in a PGM. In addition, McNultty and colleagues [34] demonstrated that psychological stress can increase the spontaneous neurotransmission recorded in a PGM. It should be remembered that the neuromuscular junction of skeletal muscle in mice is innervated by the sympathetic nervous system [35]. Based on these data, it can be suggested that the mechanism by which the CAD is capable of increasing spontaneous neurotransmission is due to an excitation of the sympathetic nervous system caused more by the accumulation of fat than by the inflammatory state of the muscle secondary to fat. On the other hand, the spontaneous neurotransmission elevated and maintained for several weeks after stopping the diet and normalizing the weight of the mice could also be due to the increased activity of the sympathetic nervous system, in this case, it is secondary to the stress suffered by the animals when being deprived of the hypercaloric diet. This last situation has been described by Lalanza and co-authors [22], who observed that by suppressing a cafeteria diet in rats, an increase in their anxiety levels is caused. In this sense, it is known that the following diets based on tasty foods rich in fat can produce an addictive effect and its suppression generates a withdrawal syndrome that partly explains this increased anxiety [36,37]. 

On the other hand, the increase in neurotransmission may be due to the development of a pro-inflammatory state in the muscle because of diet. In this sense, Khan et al. [29] demonstrated that after only 2 weeks of HFD, macrophages already appear in the muscle fat tissue of mice. Furthermore, Fink et al. [25] showed that for mice exposed to a diet rich in fat, abundant pro-inflammatory macrophages and neutrophils appear early and that it increases with time. Taken together, these data suggest that from the first weeks of exposure to high-fat diets, a pro-inflammatory state begins to develop in muscle tissue, even before significant weight gains are achieved. In other words, changes could occur in the release of ACh without accumulations of fat in the muscle, as it occurs for the mice exposed to HFD in the present study.

A common factor for the two types of diet used in this study is their high content of saturated fat. It is well known that saturated fats have a direct effect on the immune system by activating the release of pro-inflammatory cytokines [38]. In a recent study, Song et al. [26] found that subjecting rats to a diet rich in fat produced an increase in the postoperative pain in them. This fact coincides with other similar studies evaluating other types of pain models such as inflammatory or neuropathic [39,40]. In addition, Song et al. [26] demonstrated that a single week of a high-fat diet was not capable of producing obesity, but an increase in the pain response, although of lesser magnitude. That is, the increase in pain seems to be more related to the diet itself than to obesity. In addition, we also observed that the changes caused by an HFD returned faster to normality than the changes produced by cafeteria diet, which are maintained for one more week. This can be related to fat accumulation in muscle. All the changes observed between the two diets deserve more investigation and provides a model to investigate systemic vs. local effects of fat accumulation. 

Within the possible relationship between the nervous system and the increase in spontaneous neurotransmission, some authors have proposed that some of the persistent changes in obese individuals after losing weight may be due to phenomena related to synaptic neuroplasticity [40]. In the results obtained in the present study, this phenomenon could intervene in the maintenance of elevated spontaneous neurotransmission after stopping the diet.

Several studies correlate the administration of high-fat diets or cafeteria diets with being overweight and with significant increases in the pain response of different pain models: neuropathic, postoperative, and inflammatory [26,39,40]. Currently, there are no studies that assess whether this type of diet has any influence on myofascial pain. The central factor of MPS are the MTrPs and their essential characteristic is an increased spontaneous neurotransmission [18]. The present study suggests that this type of hypercaloric diets could facilitate the development of MTrPs by increasing the spontaneous neurotransmission they generate. However, it is not clear if the effects come exclusively from diet or if being overweight and accumulating fat may also play a relevant role.

## 5. Conclusions

Exposure for six weeks to a hypercaloric diet (cafeteria or highfat) causes overweight in mice, an increase in adipocytes in muscle tissues and an increase in spontaneous neuromuscular neurotransmission. Upon abandoning the diets, the mice recover their weight rapidly, but spontaneous neuromuscular neurotransmission remains elevated. In other words, the alteration of spontaneous neurotransmission is not exclusively related to being overweight or to an increase in muscle fat. Overall, it can be concluded that exposure to a hypercaloric diet for 6 weeks in mice may be a predisposing factor for the development of MPS and other muscle alterations aggravated by a maintained increase in spontaneous neuromuscular neurotransmission.

## Figures and Tables

**Figure 1 nutrients-12-03216-f001:**
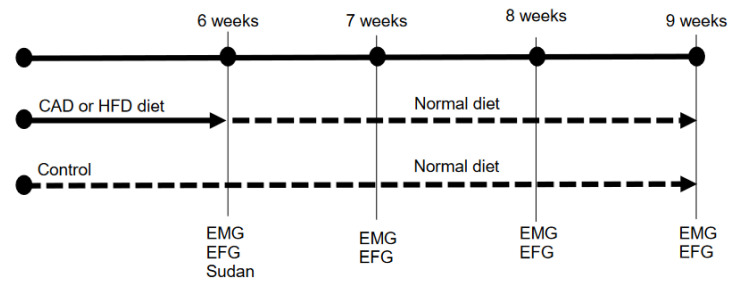
Timeline of the experimental procedure. CAD, cafeteria diet; HFD, high-fat diet; EMG, electromyography (endplate noise recordings); EFG, electrophysiology (intracellular recordings).

**Figure 2 nutrients-12-03216-f002:**
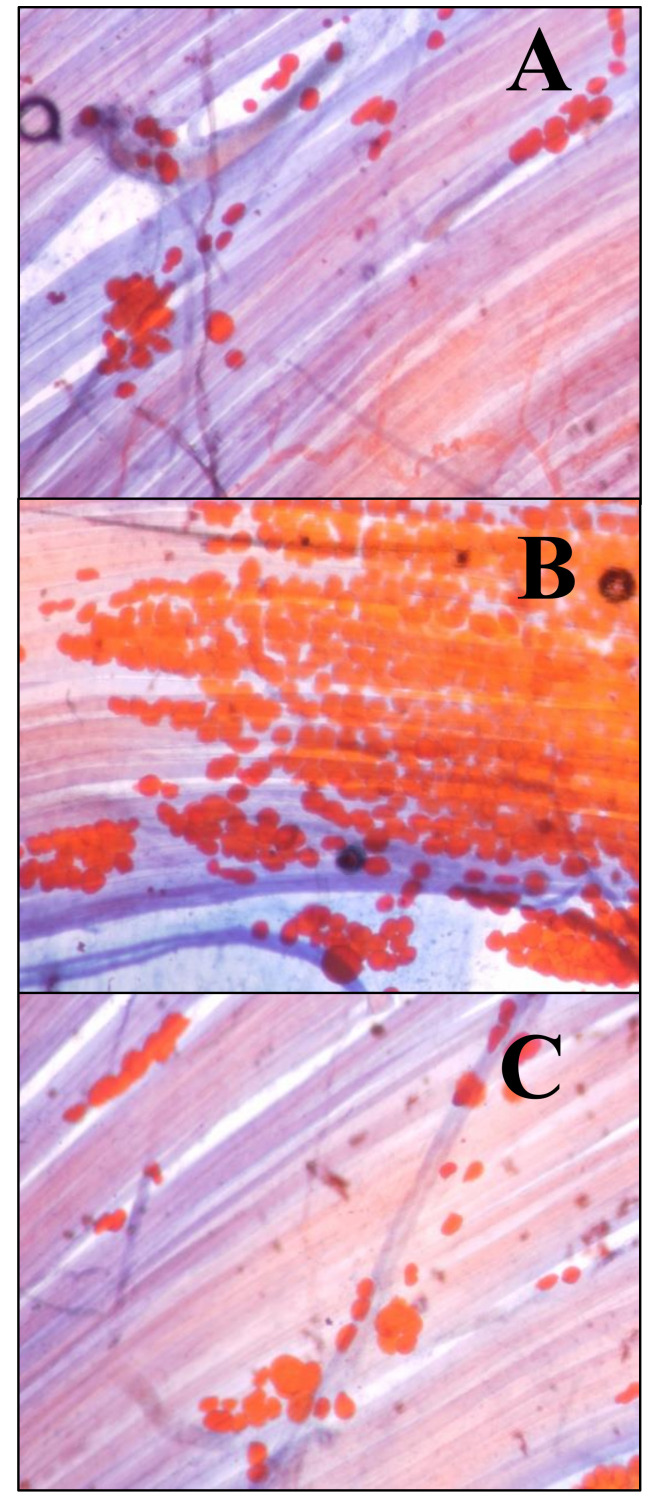
Intramuscular adipocytes. The adipocytes were stained with Sudan III. Fat looks orange. Methylene blue has been used as a contrast dye. (**A**) *Levator auris longus* (LAL) muscle from a control animal. (**B**) LAL muscle from an animal exposed to a cafeteria diet for 6 weeks. (**C**) LAL muscle from an animal exposed to a high-fat diet for 6 weeks. Initial magnification 400×.

**Figure 3 nutrients-12-03216-f003:**
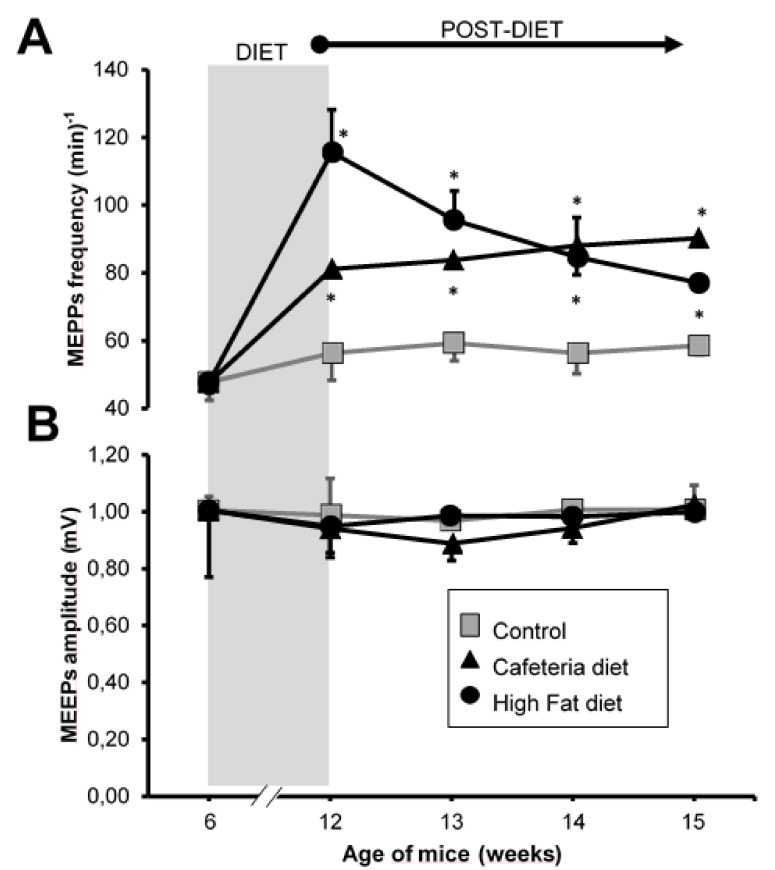
Intracellular recordings. (**A**) Frequency of miniature endplate potentials (MEPPs) expressed as number of events per minute. (**B**) Mean amplitude of the MEPPs expressed in mV. Grey area, period in which the animals were exposed to the CAD or HFD diets. Values are expressed as the mean ± SEM. For each experimental series, *N* = 4 animals. * *p* < 0.05 with respect to control values. Cafeteria diet group, CAD. High-fat diet group, HFD.

**Figure 4 nutrients-12-03216-f004:**
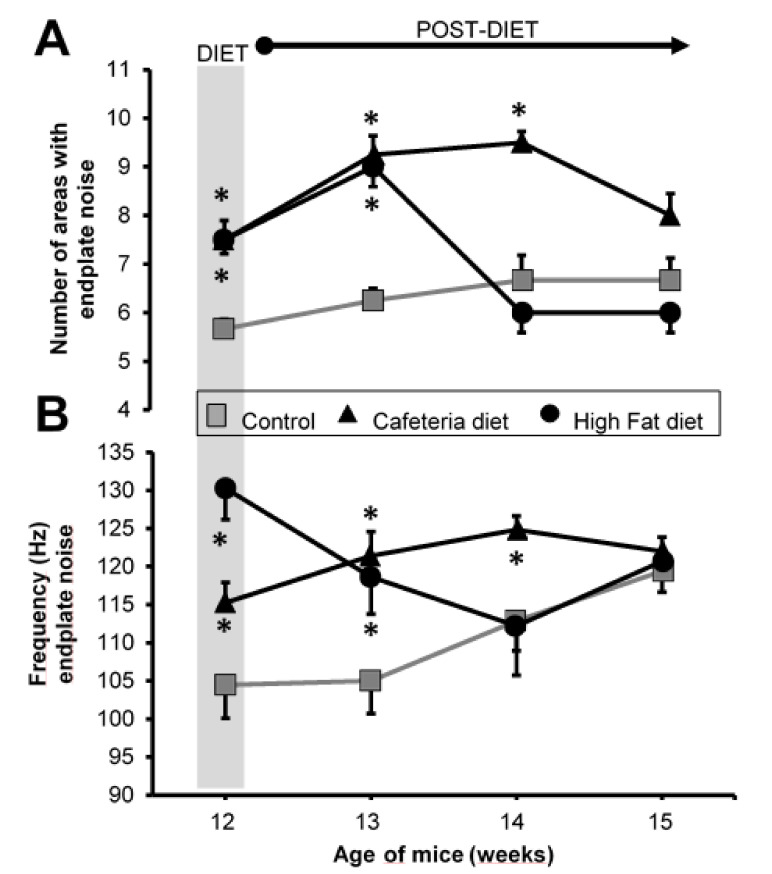
Electromyography. (**A**) Number of average areas with plate noise. (**B**) Each area with endplate noise was analyzed by quantifying the number of events/s (Hz). Grey area, end of the period in which the animals were exposed to the diets. Values are expressed as the mean ± SEM. For each experimental series, *N* = 8 gastrocnemius muscles from 4 animals. * *p* < 0.05 with respect to control values.

**Table 1 nutrients-12-03216-t001:** Nutritional facts of the diets used.

	CAD	HFD	Regular Chow Diet
Calories (kcal)	459	532	397
Total Fat	23	60.6	6.9
Saturated Fat	11	21.7	-
Total Carbohydrate	56	26.3	68
Sugars	24	9.7	-
Dietary Fiber	2.5	-	-
Protein	5.3	13.1	25
Sodium	0.65	0.23	0.3

The nutritional data are expressed per 100 g. Cafeteria diet (CAD). High-fat diet (HFD).

**Table 2 nutrients-12-03216-t002:** Weight evolution.

Procedure	Age	Control	CAD	HFD
**1**: 6 weeks of age; **2**: 6 weeks with diet	12 weeks	38.00 ± 2.85 (*n* = 20)	58.03 ± 4.04 * (*n* = 20)	55.15 ± 1.39 * (*n* = 20)
**1**: 6 weeks of age; **2**: 6 weeks with diet; **3**: 1 week without diet	13 weeks	43.73 ± 0.88 (*n* = 12)	45.53 ± 1.03 (*n* = 12)	45.11 ± 2.11 (*n* = 12)
**1**: 6 weeks of age; **2**: 6 weeks with diet; **3**: 2 weeks without diet	14 weeks	43.24 ± 0.86 (*n* = 8)	45.50 ± 0.98 (*n* = 8)	43.97 ± 3.09 (*n* = 8)
**1**: 6 weeks of age; **2**: 6 weeks with diet; **3**: 3 weeks without diet	15 weeks	42.17 ± 2.85 (*n* = 4)	44.67 ± 1.22 (*n* = 4)	43.25 ± 2.57 (*n* = 4)

Values are expressed as the mean ± SEM. Cafeteria diet group, CAD. High-fat diet group, HFD. *, *p* < 0.05 with respect to the weight of the control animals.

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
