# Peer review of "Effects of a Fat-Rich Diet on the Spontaneous Release of Acetylcholine in the Neuromuscular Junction of Mice"

_nutrients, 2020, doi:10.3390/nu12103216_

Round 1

Reviewer 1 Report

The authors need to mention how many rats were used for each experiment in each experimental sections. Originally they started with n=20 in each group.

The authors need to use references for each of the methodology presented in the study?

Line 121:Dietary manipulations, the authors may provide detailed diet composition in a table.

All of the figures with X axis shows the age of the experimental rats, which is very confusing? because the experimental period is only for 9 weeks? need to revise all of the figures.

Line 165: Figure 2 legend "∗, P < 0.05 with respect to week 6 weight", need some clarification, significance can be evaluated using a control group?

Line 224: Figure 3 legend, the abbreviations "LAL" can be mentioned

Line 235: Sudan II staining  reframe/need to improve on the method "After washing off the excess under running water" how did the authors keep the muscle under running water?

Line 246: (see figure 2) to be read as (Figure 2) need to be changed.

Figure 5, Y axis "Frequency edplate noise" check the spelling.

Reviewer 2 Report

Dear Dr. Santafe & colleagues,

Thank you for the opportunity to review your manuscript. Please find below a synopsis of my thoughts and comments:

Major

My greatest concern regarding your manuscript is that your statistical approach is not appropriate for the study design. You state that Welch’s t-tests were used. However, repeated-measures analyses of variance (ANOVA’s) would have been more appropriate as the same variable was repeatedly measured over a number of weeks in the same study population (i.e. body mass, MEPPS, endplate noise and frequency). As it appears multiple t-tests have been performed to test between groups at different time points, it would seem that type I error has been evoked.

The flow and structure of your methods section require attention. There are some points within the methodology where sections in the latter parts of the methods would be better explained earlier on. For example, section 2.7 would be better incorporated into section 2.4 where you first introduce Sudan III fat staining. In my opinion, section 2.1 would be better renamed “Animals and Dietary Protocol” with lines 97-105 and section 2.2 incorporated accordingly.

If you wish to keep the information on lines 115-120, a condensed version of this would be better served at the end of your introduction on line 76 as this will help contextualise your key findings provided on lines 76-79.

Lines 229-236 – On line 147, you state that both the LAL and gastrocnemius were used for Sudan III staining. Yet, there is no mention of the gastrocnemius in section 2.7, nor are there any data within the results pertaining to adipose tissue within the gastrocnemius. Is this simply an error, or is there data pertaining to this muscle, as a comparison between two muscles of different anatomical locations would further strengthen the paper.

I am surprised that no quantifiable approach to measuring muscle fat was used, via ImageJ or a similar software. The images in figure 3 may be selected on bias to demonstrate a best case and worst case scenario, when quantifying fat content following the Sudan III staining in different animals and in different areas of the muscle would have been far more robust. The results in section 3.2 (lines 292-294) are therefore effectively meaningless.

How was dietary adherence checked? It does not appear that measured body mass of each animal was recorded during the dietary protocol. Moreover, a check of adiposity beyond animal body mass would have been beneficial, either performed in vivo in the form of Body Mass Index, Lee Index of Obesity or waist circumference, or posthumously via cadaveric excavation of subcutaneous fat. I state this because figure 2 and section 3.1 are difficult to follow.

The explanation of the change in animal body mass is highly confusing. Figure 2 does not include the control data, nor the animal body masses at the start of the dietary protocol, despite stating “control, N=20” in the figure legend. Therefore, your purported claim that the HFD and CAD diets causes body mass to be higher than control cannot be verified by the figures. The symbols indicating significant in figure 2 are therefore pointless as all it shows is the HFD & CAD animals increased their body weight against 6 weeks, but you state the same was true of control animals. It would be far easier to report mean animal body mass with standard deviation, perhaps in a table, and let the readers see the data for themselves.

I find it extremely hard to believe that animal body mass would recover to pre-diet values within the space of one week (week 12-13 as per figure 2).

Minor

Line 51 – “TNF-a” should be “TNF-α” (i.e. alpha) and should be corrected throughout the manuscript.

Line 57 – It might be worth stating that MTrPs is a spontaneous phenomenon.

Line 58 – The prevalence of MTrPs is in what population?

Line 61 – “clef” should be “cleft”.

Line 63 – I don’t understand what happens to local acid pH. Does this decline in response to an MTrPs event?

Line 115 – 120 – This information feels superfluous as it is explained in full in the later sections. For example, the detailed description of the muscle experiments are in section 4, so I don’t think it is necessary to state this in section 2.1.

Line 115 – The LAL muscle has yet to be defined, so should be stated in full as the levator auris longus muscle. Moreover, is there a rationale as to why this muscle in particular was selected for histological analyses when locomotor muscles (e.g; soleus/EDL) are commonly used for such analyses?

Line 119 – EFG should be stated in full at the first use.

Section 2.2 – The hyphens at the start of each paragraph are unorthodox.

Line 131-132 – If animals are housed in cages of 2, one animal may consume more food than the other. Therefore, it would be more appropriate to say each cage of 2 animals consumed 28g of the CAD diet per week.

Line 145 – I know that Ringer solution is standard, but the millimolar concentrations of the compounds in the Ringer solution should still be stated. Moreover, the “2” in O2 and CO2  should be subscripted.

Line 181 – The abbreviation of membrane potential is rudimentary as it is not used later in the manuscript.

Lines 229-236 – On line 147, you state that both the LAL and gastrocnemius were used for Sudan III staining. Yet, there is no mention of the gastrocnemius in section 2.7, nor are there any data within the results pertaining to adipose tissue within the gastrocnemius.

Figures

Figure 2 – Why not just show the mean body mass data. For each dietary group at each time point, including week 6.

Figures 4 & 5 - The y-axis in figure 4 & 5 have spelling mistakes (MEEPs should be MEPPS; edplate should be endplate). Additionally, each panel in each figure would be better labelled A & B rather than “top” or “bottom”. I would also suggest using a 2break” in the x-axis between 6 and 13 weeks as the x-axis currently illustrates that the time scale is equal between each point.

Reviewer 3 Report

The present study aimed to assess whether obesity or overweight may be a facilitator of myofascial pain. The authors conducted the study on male Swiss mice. The authors found the supply of a hypercaloric diet for six weeks in mice increases spontaneous neurotransmission, thus facilitating the development of MTrPs. This is an interesting study. I have some minor comments that hope the authors can clarify.

First, why did the authors only choose male mice? Is there any concern of their sex?

Second, in Figure 2, did the “age” on the X axis indicate the study period or the age of the animals?

Third, how did the authors compare the values among several groups simply by using T test? Did they adjust their p value to avoid the bias of multiple comparisons? Have they considered to use non-parametric ANOVA to deal with this issue?

Round 2

Reviewer 2 Report

Thank you for taking the time to adequately address my concerns. I have no further comments.

Reviewer 3 Report

Good work!